# Forgotten, But Not Lost—Alloparental Behavior and Pup–Adult Interactions in Companion Dogs

**DOI:** 10.3390/ani9121011

**Published:** 2019-11-21

**Authors:** Péter Pongrácz, Sára S. Sztruhala

**Affiliations:** Department of Ethology, Eötvös Loránd University, Pázmány Péter sétány 1/c, 1117 Budapest, Hungary; sztruhalasara@hotmail.com

**Keywords:** behavior, dog, alloparental care, puppies, breeders

## Abstract

**Simple Summary:**

Companion dogs are vastly popular animals; however, we know surprisingly little about their natural parental behaviors. Meanwhile, although wolves, dingoes, and, to an extent, even free-ranging dogs show several forms of alloparental behaviors, the parental care among companion dogs is thought to be solely provided by the mother. We circulated an international survey for dog breeders, asking them about the forms of alloparental behaviors they observed among their dogs, as well as further interactions between the puppies and other adult dogs at home. Our results show that allonursing and feeding of the pups by regurgitation is a widespread phenomenon among companion dogs. The behavior of young puppies regarding, for example, their reaction to other dogs’ barking was also influenced by the timing of their access to the other dogs at the breeder’s home. Based on the breeders’ observations, sexual status and age of the other dogs affected the way they interacted with the puppies, and also the way the puppies’ mother interacted with them. These results highlight the importance of dog–puppy interactions during the early weeks of life, an often neglected area compared to the well-known elements of puppy socialization with human beings.

**Abstract:**

Socialization with humans is known to be a pivotal factor in the development of appropriate adult dog behavior, but the role and extent of dog–dog interactions in the first two months of life is rarely studied. Although various forms of alloparental behaviors are described in the case of wild-living canids, the social network of companion dogs around home-raised puppies is almost unknown. An international online survey of companion dog breeders was conducted, asking about the interactions of other dogs in the household with the puppies and the pups’ mother. Based on the observations of these breeders, our study showed an intricate network of interactions among adult dogs and puppies below the age of weaning. Alloparental behaviors (including suckling and feeding by regurgitation) were reportedly common. Independent of their sex, other household dogs mostly behaved in an amicable way with the puppies, and in the case of unseparated housing, the puppies reacted with lower fear to the barks of the others. Parousness, sexual status, and age of the adult dogs had an association with how interested the dogs were in interacting with the puppies, and also with how the mother reacted to the other dogs. Our study highlights the possible importance of dog–dog interactions during the early life of puppies in forming stable and low-stress interactions with other dogs later in life.

## 1. Introduction

The species-specific traits of dogs (*Canis familiaris*) are mostly inseparable from the process of domestication. The evolutionary changes in the socio-cognitive capacities [1,2], behavior [3,4], anatomy and physical appearance [5,6], as well as the physiology [7,8], are most often evaluated with regard to the differences between dogs and their closer or more distant wild-living relatives. More recently, researchers concentrated on the behavioral and cognitive features [9,10], both as proximate and ultimate factors behind the adaptation of dogs to the anthropogenic niche, with only a few exceptions—for example, genetic changes that could affect the carbohydrate metabolism in dogs were also highlighted as assumed key factors behind domestication [11,12]. Although differences in the reproductive biology of dogs (e.g., switching from being monestrous to a mainly diestrous cycle), when compared to their closest wild relative, the grey wolf (*Canis lupus*), are also apparent among the crucial changes, and have also been modeled by the well-known silver fox project conducted in Novosibirsk, Russia [13], the reproductive behavior of companion (or “family”) dogs is rarely discussed in scientific literature, apart from various issues covered by veterinary science (e.g., [14,15]). As the reproduction of companion (and working) dogs is mainly planned, supervised, and restricted by human caretakers [16,17], this segment of dog behavior remains almost untouched by ethologists. Furthermore, alloparental behavior and paternal caretaking of the young, two factors which are considered uniquely typical for a wide selection of canid species [18], are literally unknown among companion dogs, and have only recently been discovered in the free-ranging dog populations [19,20].

Free-ranging dogs are often considered to be the “ecologically most successful” variants of domestic dogs due to their vast number (according to some estimations, around 800 million worldwide—[21]) and ubiquitous presence in and around human settlements. Subsequently, it is assumed that free-ranging dogs provide the best opportunity to understand the biology of dogs [21], as free-ranging dogs have been adapting to their environment for many generations without excessive artificial selection by humans. However, when it comes to parental behavior, seemingly, there is a considerable difference, even among free-ranging dogs. This has led to such widely differing observations, which have either stated that lack of alloparental care is one of the reasons why mortality rate of young pups is very high among free-ranging dogs (in Italy—[22]; in Mexico—[23]), or described more or less sporadic, but existing paternal/alloparental caretaking (in India—[20,24]). In their exhaustive review on canid reproduction, Lord and colleagues [25] assumed that since domestic dogs became dependent on human resources, they mostly lost the need for alloparental caretaking (i.e., because they have a stable food supply); meanwhile, the same ecologically predictable food resources made it possible that the sexual behavior of dogs also mostly lost the strict seasonality that is typical to wild canids. This feeding ecology-based theory gains further (indirect) support from the observations made with dingoes—feral dogs in Australia that became isolated from other Southeastern Asian dog populations around 3.5–5 thousand years ago [26]. These dogs sustain themselves mainly by hunting large prey—consequently, they have retained many typical features of the reproductive behavior of wolves (e.g., alloparental care—[27], monestrus—[18]), because these seem to support the lifestyle of apex canid predators.

Very little is known about the natural interactions of juvenile (pre- and around weaning period) dogs and their older canine companions (kin and non-kin) in the case of companion (pet) and working dogs. Contrary to dog–human interactions during puppyhood, which were recently investigated from multiple aspects and considered to be a crucial part of the “process of proper socialization” [3,28,29,30], the behavior and effect of adult dogs on puppies in the home environment have received much less interest from investigators. Among the most likely reasons for this is the difficulty of conducting observations at the owners’ home, or the highly variable social environment (i.e., there is no standardized or “natural” social structure at breeders’ homes that would include roughly the same kinds of adult dogs around the puppies). Consequently, although there are data about the interactions of dog puppies with other dogs at public areas [31], as well as pup–pup interactions within the litter (e.g., the ontogeny of playful behavior [32]), our knowledge of pups’ interactions with familiar, but not necessarily related adult dogs from the household is very limited. The exception is the interaction with the mother—as the extent and style of maternal care was found to have fundamental effects on later behavior in working dogs (e.g., drug seeking dogs—[33]; police dogs—[34]). However, except for the work of [35] regarding the feeding of the pups by regurgitation, we do not know of any studies on alloparental behavior within pet dog groups, and indications of paternal care are missing as well.

In this study, we conducted a detailed, international internet questionnaire about alloparental behavior and pup–adult dog interactions within companion dog groups that live at the homes of dog breeders. We surveyed not only the existence of alloparental nursing and feeding of pups by regurgitation, but we also covered such behaviors as the mother dog’s reaction to other adult dogs around her offspring, and the pups’ reaction to other household dogs’ barking. We analyzed whether the aforementioned behavioral variables were dependent on the circumstances of how the pups were kept—especially with regard to their isolation from the other dogs in the home. Based on the literature about the reproductive biology of free-ranging dogs [36] and dingoes (e.g., [18]) (both of which are not under direct human control), it can be assumed that alloparental behavior emerges in dogs as a functional response of the temporal predictability and ease of access to food [25]. As companion and working dogs in private or professional kennels are steadily provided with easily accessible food, one could assume that the need for alloparental and paternal care is minimal to non-existent. However, as an alternative hypothesis, one could expect that even companion dog populations have retained the capacity of providing alloparental care, as we do not know about active selection that would go against this capacity in dogs under human management.

## 2. Materials and Methods

### 2.1. Ethical Approval

This study was carried out in accordance with national and international ethical guidelines (e.g., American Psychological Association, Hungarian Psychological Association). Participation was voluntary; we handled all data obtained confidentially, and anonymized the questionnaires after data collection. The Ethical Committee of Eötvös Loránd University reviewed and approved the study. Ethical permission number: PEI/2016/003.

### 2.2. Development of the Survey

Two questionnaires were created in Hungarian and English languages, and both versions were disseminated via social media and email. Questionnaire 1 (https://goo.gl/forms/u7Q4ti2jglUHfKy63) was the main endeavor with a complex set of items, while Questionnaire 2 (https://forms.gle/ceRXWfrw4tEmtAUZ6) served to collect additional information regarding adult dogs’ food regurgitation to puppies. It took approximately 10–15 min to complete Questionnaire 1, and 5 min to complete Questionnaire 2. Data from Questionnaire 1 were recorded between October 2017 and April 2018. Questionnaire 2 (focusing on solely the regurgitating behavior) was circulated between 8–19 of October 2018.

Nonprobability convenience sampling was used, as the questionnaires were distributed via social media, predominantly by sharing them in various Facebook groups, dedicated to the breeding of specific dog breeds, or breeding of purebred dogs in general. We repeated the call for participation weekly on the social media platforms along the course of the survey. Additionally, the surveys were also sent via email to dog breeders on the basis of personal acquaintances of the authors—however, this resulted in only about 10% of the total sample.

### 2.3. Subjects

Participation in the survey was voluntary and anonymous; however, participants could provide their name and/or contact address on a non-mandatory basis. No form of incentive was offered for the participation. 

Our first sample consisted of 77 dog breeders from 11 countries, who reported their observations of 45 dog breeds. Our second sample from Questionnaire 2 consisted of the observations of 36 dog breeders from 3 countries and 28 dog breeds (see Table 1).

Breeders who completed the questionnaire had to meet the following criteria: They raised at least one litter of puppies a priori the completion of the questionnaire; they keep at home a minimum of one more dog of any age, in addition to the mother of the puppies. We requested that the breeders answer the questions regarding their observations on their “entire experience” of their past litters bred.

### 2.4. Variables

The questionnaires covered the first 8 to 12 weeks of the puppies lives which are spent in their breeders’ home. During this period, the puppies normally experience various new stimuli (both social and asocial). Towards the end of this period, puppies start to show an almost complete set of social behaviors, including playful, agonistic, and communicative interactions with not only their kin [37], but also towards humans [38].

Beyond the demographic details, the questionnaires contained items that were aimed at the following interactions between the mother, other dogs, and the puppies:-Alloparental behaviors (apart from feeding by regurgitation); e.g., when another female dog nursed, or attempted to nurse the puppies. (Multiple choice, where the possible answers were: No alloparental behaviors were observed; puppies were nursed by—the daughter, the mother, the sister of the mother dog; an unrelated adult female nursed the puppies.)-Feeding with regurgitation (we asked the participants to indicate separately whether the mother of the puppies and/or another dog regurgitated food for the puppies). Multiple choice item.-Puppies’ reaction to adult dogs’ barking: Breeders could select one of three options: Puppies show no reaction, puppies become frightened (run and hide, become quiet), or puppies start to bark when they hear the adults’ barking.-Adult dogs’ reaction to the puppies’ whining: We again used a three grade scale (single choice): Adults show no reaction to puppies’ whining, they sniff the puppies, or the adults show alloparental behavior (mother-like caretaking behavior towards the puppies: Licks them, lays down next to them, plays with them).-Adult dogs’ reaction to puppies in general situations: During play, eating, drinking, and chewing on toys (single choice). We got the majority of answers from the following two categories: Adults show no reaction to the puppies, or adults are friendly with the puppies. Other categories, such as adults start barking, show aggression towards the puppies, puppies show aggression towards adults, were excluded due to the very sporadic or missing answers to these options.-Mother dogs’ reaction to adult dogs around her puppies (single choice). Breeders could again indicate their answers on a three-grade scale: She shows no reaction, she is aggressive with other dogs, or she is friendly with other dogs.

We analyzed the possible associations among the aforementioned parameters and the following factors:-The way the puppies are separated from the other dogs. Three alternatives were given: Puppies and adult dogs are kept together, puppies are separated with a fence or barrier, or puppies are separated completely from adult dogs. (i.e., they live in a separate room of the house).-Age of the puppies when they are separated from adult dogs. Participants could choose from three main options: The puppies and adult dogs are kept together, the puppies are kept separated only until a defined age, or the puppies are separated completely from other adult dogs the entire time they are at the breeder.-Aggression towards puppies. The breeders were asked whether they experienced any aggression towards the puppies in general situations (yes/no).

In the case of the items that included the puppies’ or their mother’s interaction with “another dog”, there were separate options for the following categories of dogs:-The age of the other dog: Younger than one year, between one and eight years, and over eight years old.-Paternal status of the other dogs (whether it was the father or not of the current puppies).-Previous parental experience of adult dogs (i.e., has the dog already reproduced, in the case of both male and female dogs); yes or no answers could be selected.-Sexual status of the adult dogs: Breeders could select whether the particular dog was neutered/spayed or intact.

The following categories were used as fixed factors in the statistical analysis.

Dog breeds categorized according to the FCI (Fédération Cynologique Internationale) system. For the statistical analysis, we had a large enough sample size from only three FCI breed groups: Group 1 “Sheepdogs and Cattledogs”, Group 2 “Pinscher and Schnauzer—Molossoid, Swiss Mountain, and Cattledogs”, and Group 9 “Companion and Toy Dogs”. 

Dog breeds were categorized according to their genetic distance from the wolf (*Canis lupus*) (based on [39]). Three out of the ten groups had a large enough sample size for statistical analysis: “Working Dogs”, “Herding Dogs”, and “Mastiffs”.

### 2.5. Data Analysis

For the statistical analysis, we used the IBM SPSS statistical program (version 22.0, Armonk, NY, USA). Raw data are available online as a Appendix A.

The alloparental behavior was handled as a binary variable (presence/absence) and its association with the fixed factors was analyzed by generalized linear models (GzLM) with the binary logistic method. The same method was used when we analyzed the adult dogs’ behavior to puppies in general situations. The binomial test was used to analyze the occurrence of regurgitating behavior (presence/absence). The puppies’ reaction to adult dogs’ barking was analyzed by ordinal regression. We used GzLM with ordinal logistic to find out whether there were differences in the adult dogs’ behavior towards the puppies’ whining and in the mother’s reaction towards the adult dogs. We used sex, previous parental experience, sexual status, and age of the adult dogs as independent variables. 

In the case of models where there were two or more independent variables, the two-way interactions were also included in the analysis. Alpha was set at 0.05 in each test.

## 3. Results

### 3.1. Alloparental Behaviors (Including Nursing, Licking, Cleaning, But Not Regurgitation) 

From the 77 responses to Questionnaire 1, we found in 61% of the cases that breeders observed the presence of alloparental nursing behaviors. We found no significant association between the adult dogs’ caretaking behavior and any of the fixed factors (GzLM with binary logistic—housing method of puppies κ^2^(2) = 3.022, *p* = 0.221; timing of separation of the puppies κ^2^(2) = 2.349, *p* = 0.309; aggressive behavior with the puppies κ^2^(1) = 0.245, *p* = 0.621; FCI breed groups κ^2^(2) = 0.294, *p* = 0.863; and genetically clustered breed groups κ^2^(2) = 0.539, *p* = 0.764).

### 3.2. Feeding of the Puppies with Regurgitation

From the 41 breeders who responded to Questionnaire 2, 34 reported that he/she observed mother dogs regurgitating to their puppies. By setting the chance level to 0.5, according to the binomial test, this ratio is significantly above chance level (*p* < 0.001). However, one could also expect that mothers will almost always feed their litters by regurgitation (parallel with the pariah dogs, [36]), therefore we ran the binomial test again with the chance level set at 0.99. The abovementioned observed ratio is significantly below this (*p* < 0.001). We also investigated the occurrence of alloparental regurgitative feeding. From the 41 responses, 18 breeders reported observations of other (i.e., not the mother) dogs providing food by regurgitation to the puppies. Based on the very sporadic incidence of this behavior, based on the literature about free-ranging dogs [20,36], we set the reference ratio near zero (0.01). The observed ratio was significantly higher than this (binomial test, *p* < 0.001).

### 3.3. The Puppies’ Reaction to the Other Dogs’ Barking

We found a significant association with the housing method of puppies (ordinal regression—κ^2^(2) = 9.363, *p* = 0.009). Based on the post-hoc comparisons (Table 2), puppies show the weakest reaction to the barking of adult dogs when they are not separated from the other dogs; meanwhile, they show fear or they join in barking with the others when they are kept partly or fully separated from the other dogs in the household (Figure 1). A similar, but non-significant trend was found in the case of the timing of separation of puppies (ordinal regression—κ^2^(2) = 4.631, *p* = 0.099), where again, puppies that are never separated from the other dogs showed the weakest reaction to other dogs’ barking according to the breeders’ observations. We found no further significant associations between reaction to other dogs’ barking and the fixed factors (ordinal regression—aggressive behavior with the puppies κ^2^(1) = 0.628, *p* = 0.428; FCI breed groups κ^2^(2) = 1.112, *p* = 0.573; and genetically clustered breed groups κ^2^(2) = 0.598, *p* = 0.742).

### 3.4. Adult Dogs’ Reaction to the Puppies’ Whining

We found no difference between the reaction of the puppies’ father and other adult male dogs (GzLM with ordinal logistic—κ^2^(1) = 0.606, *p* = 0.436). In the case of the adult female dogs’ age, reproductive status (intact vs. spayed), and parousness (had offspring vs. did not have offspring previously), we found a significant association between the age of the female dogs and their reaction to the puppies’ whining (GzLM with ordinal logistic—age κ^2^(2) = 8.564, *p* = 0.014). According to the post-hoc comparisons (Table 3), female dogs under 1 year of age showed the strongest reaction (including either sniffing, or even more intense, mother-like behaviors, such as licking, lying beside, or attempting to nurse) to the puppies’ whining (Figure 2).

Neither the reproductive status (κ^2^(1) = 2.442, *p* = 0.118) nor the parousness (κ^2^(1) = 2.341, *p* = 0.126) of the adult females had significant association with reaction to the puppies’ whining. We did not find any significant interactions between the factors. According to the breeders’ observations, the parousness of the adult dogs (including both males and females) had a significant association with their reaction to the puppies’ whining (Table 4; GzLM with ordinal logistic—κ^2^(1) = 5.795, *p* = 0.016), where parous adult dogs react more intensely than the nulliparous ones (Figure 3). Sex itself did not have a significant effect (κ^2^(1) = 2.240, *p* = 0.134), and we did not find significant interaction between the factors.

### 3.5. Adult Dogs’ Reaction to Puppies in General Situations

Adult males’ reactions did not show a difference between the puppies’ father and other males (GzLM with binary logistic—κ^2^(1) = 0.000, *p* = 1.000). The age (GzLM with binary logistic—κ^2^(2) = 8.300, *p* = 0.016) and reproductive status (κ^2^(1) = 6.293, *p* = 0.012) of female dogs showed a significant association with their reaction to the puppies—according to the post-hoc comparisons (Table 5), less than 1 year old females and the intact females were more likely to react with playful interest to the puppies’ presence than the older or spayed females (Figure 4). The parousness of the females did not have a significant effect on this parameter (κ^2^(1) = 0.271, *p* = 0.603). The sex (κ^2^(1) = 0.451, *p* = 0.502) and parousness (κ^2^(1) = 0.164, *p* = 0.686) of the other adult dogs in the household, as well as the interaction of these factors, did not show a significant association with the reaction to the puppies in general encounters.

### 3.6. The Reaction of the Mother of the Puppies to Other Dogs in the Household

In the case of other adult male dogs, we did not find a significant difference between the reaction of the mother to the father of the puppies or other adult male dogs (GzLM with ordinal logistic—κ^2^(1) = 1.091, *p* = 0.296). When we analyzed the reaction of the mother to other females in the household, according to the breeders’ observations, there was a significant association with the other females’ age (GzLM with ordinal logistic—κ^2^(2) = 13.090, *p* = 0.001), and we also found a significant interaction between the reproductive status and parousness of the other females (κ^2^(1) = 12.183, *p* < 0.001). According to the post-hoc comparisons, mother dogs are less friendly with other adult females than with juveniles or older females. Furthermore, mother dogs are the least friendly with nulliparous, intact females (Table 6). This result was strengthened by another analysis, where we tested the association with the sex and parousness of other dogs in the household. Besides the two significant main effects (GzLM with ordinal logistic—sex κ^2^(1) = 12.784, *p* < 0.001; parousness κ^2^(1) = 15.090, *p* < 0.001), we also found a significant interaction (κ^2^(1) = 5.732, *p* = 0.017). According to this, mother dogs are the least friendly with nulliparous females by the observations of the breeders.

## 4. Discussion

In this study, we surveyed an international sample of active dog breeders, with questions aimed at the various forms of alloparental caretaking of the puppies, the interactions between puppies and other dogs apart from the mother, plus the interactions between the mother dog and other dogs at the same home. Based on the breeders’ reports, alloparental nursing can be considered widespread in companion dogs, although its presence is significantly less than 100%. Other forms of nurturing the puppies (i.e., regurgitation) is typical for the mothers, and additionally provided by other dogs in slightly less than 50% of the cases as well. The presence of alloparental caretaking was not associated with the type of breed, or with the way the puppies were separated (or not) from the other dogs. The puppies showed a stronger reaction to other dogs’ barking if they were kept at least partially separated from other dogs in the household. Paternity (including both the father of the puppies or an unrelated male) did not have significant association with the dogs’ reaction to the puppies’ whining and with the dogs’ playful behavior with the puppies. Mother dogs also reacted similarly with the father of the puppies and other males. Young female, parous dogs showed stronger interest towards the whining of the puppies; and the young/sexually intact females behaved more playfully with them. In turn, mother dogs were reported as being the least friendly with other females that were either adult, or intact, nulliparous ones.

Before the detailed discussion of the results, we should consider some limitations of our study. As it was a questionnaire survey, we entirely relied on the expertise and experiences of the breeders who reported on the behavior of their dogs. Therefore, in the future, targeted behavioral tests would be useful for validating the key findings about the various dog–dog interactions—however, we should also keep in mind that conducting experiments in the homes of dog owners/breeders has its difficulties/limitations, too. The sample size was relatively low in our study, partly caused by the rather strict rules of participation (only breeders were invited with actual experience of raising at least one litter of puppies in the past with other adult dogs around). In any case, a more comprehensive survey would be beneficial (preferably involving more participants from non-European countries), especially which includes a more complete set of dog breeds from preferably each breed groups based both on the artificial (FCI) and genetic clustering. Finally, the scope of our survey was limited by the way in which the questionnaire was distributed—the utilization of internet and social media has its benefits (one can reach participants quickly and technically with no geographic limitations); however, potentially knowledgeable participants can also be left out because they do not use these means of communication.

Compared to the detailed comparative work done on the parental behavior of wild canids [25], and even on free-ranging dogs [36], empirical studies are noticeably lacking for companion dogs. Companion and working/service dogs are usually bred with close human supervision (e.g., [40,41]); therefore, the circumstances and, many times, even the process of parental care can be considered as more or less artificial compared to wild dogs. Apart from veterinary and animal breeding texts (e.g., [17,42]), scientifically accumulated information is surprisingly sparse regarding the maternal (see review [43]) and alloparental behaviors, as well as the interaction between the puppies and other adult dogs, in the first two months of life. 

Our results show that alloparental caretaking behaviors (allonursing, regurgitating of food) are widespread among dogs that are kept by hobby breeders. Fostering of the young has a solid ecological basis in such species as the super-social members of the *Canidae* family (gray wolf; African wild dog *Lycaon pictus*; dhole *Cuon alpinus*). Among others, Riedman [44] lists the following factors that, along the course of evolution, could facilitate alloparental caretaking to develop: “(1) Prolonged or energetically intensive parental investment; (2) small groups with tight kinship bonds; (3) highly social or cooperative group structure; and (4) young that are raised in high density breeding colonies”. From these, the first three conditions are typically true of the abovementioned wild canids—however, they are harder to interpret in the case of companion, or even-free ranging dogs. If we consider the latter (pariah, or village dogs) as the most valid ecotype of domestic dogs [21], we should notice that the change in feeding ecology (being mainly a scavenger instead of a hunter of large prey, [45]) could be the main driving force behind the alteration of reproductive and parental behavior in the majority of dogs. Scavenging does not require adults to act cooperatively—neither during the hunt, nor when provisioning the lactating mother and the young. Furthermore, in free-ranging dogs, the freshly weaned juveniles become mostly a competitor for the adults [21]; therefore, their role as “helpers” for the next generation is limited or non-existent. Interestingly, our results, as well as in the earlier study of [35], show a considerably common occurrence of alloparental behaviors among companion dogs. The apparent difference compared to the infrequently observed foster-behaviors among free-ranging dogs [19,20] could be the result of the complex effect of different levels of food competition, human intervention, and, in part, the density of animals around the breeding mother/puppies (see condition #iv by Riedman [44]). Based on this, we can hypothesize that the capacity for alloparental caretaking is steadily present in the domestic dog, even in the population of pets and working dogs where the conditions for both the breeding and caretaking of the young are highly artificial. The competition for resources is rather strong among free-ranging dogs [46], thus alloparental behaviors may be less adaptive (given that scavenging is the main type of food procurement). In companion dogs, alloparental behavioral tendencies are rarely discouraged—according to our results, its occurrence was not affected by the method of puppy-raising that the breeders chose. Somewhat surprisingly, we did not find a breed effect; however, this could be explained by the low sample size of individual breed groups. One could hypothesize that those breeds that are more closely related to wolves would show stronger alloparental tendencies. Although basal (or “ancient”) breeds were not represented well enough in our sample, the fact that we did not find a difference between the occurrence of alloparental behaviors among breed groups, with such widely varying functions as herding vs. toy/companion or belonging to the mastiffs vs. herding dogs, it shows that alloparental behaviors were probably less affected by recent functional selection of dogs. 

Keeping conditions reportedly affected the puppies’ reactivity to other dogs’ barking—the breeders observed the weakest reaction (i.e., lower levels of fearfulness, or tendency to join the others barking) in those puppies that were kept together with the other dogs in the household without any restriction. Although the first couple of months are considered crucial in the proper socialization of young dogs [47], and successful socialization is unequivocally considered a key factor in avoiding problem behaviors in dogs (e.g., [48]), still, the majority of scientific studies concentrate on the events of socialization that typically follow the puppies’ departure from the breeder’s home (e.g., [29,30]). Relatively few papers target the early interactions between the living environment and puppies still with their mother (e.g., [49]). In a few earlier studies (such as [50]), it was shown that early isolation of the puppies resulted in a decreased level of interactivity later with conspecifics. However, most studies have focused on dog–human interactions (e.g., [3,51]), because, in the case of companion and working dogs, the main measure of their success as adults depends on their fit to the human environment. Therefore, the observations reported in our study have a relevance from the aspect that the chance to interact uninterruptedly with other dogs from a very early age can improve young dogs’ behavior regarding dog–dog interactions. It is important to see, however, that the familiarity between the young puppies and the other adult dogs they interact with also can play a crucial role in the behavioral development of the juvenile dogs. Earlier, it was found [52] that the younger the puppy was when its new owners introduced it to other dogs, the higher the chance became that later the dog showed undesired aggression towards conspecifics. Adding these findings to our results, one could conclude that the predominantly amicable interactions (including alloparental care) with familiar dogs at the breeder’s home could provide the best experience for young puppies with their conspecifics; meanwhile, owners should be careful with the early exposure of the puppies to possibly negative experiences with unknown dogs at public areas [52]. So far, this particular aspect of environmental effects on behavioral ontogeny has only been marginally covered by other studies (e.g., [33,53]), where the focus has mainly been on other behaviors, such as interactions with the physical environment or with humans. 

We did not find a difference between the behavior of adult male dogs with the puppies, whether the particular dog was the father of the puppies or not, and the mother dog’s reaction to the males was also independent from the paternal status of the males. This result is in parallel with the reports on free-ranging dogs, where sometimes, more than one male dog was observed to be loosely associated with particular litters [36].

Friendly interactions were reportedly facilitated with the puppies if the other dog was a young, sexually intact female. Sexual status was also important in the case of the other dogs’ reaction to the whining of the puppies—besides the young females, intact and parous dogs showed stronger interest towards whining puppies. Interestingly, in a recent laboratory study, Lehoczki et al. [54] found no association between the sexual status, parousness, or sex of adult dogs and their responsiveness to playbacks of separation calls of puppies. Besides the option that breeders may misinterpret their dogs’ behavior, it is also possible that dogs behave differently in an artificial laboratory setting compared to a realistic situation at home when puppies are truly present. In the case of highly social canid species, fostering (alloparental helping) behaviors are facilitated by several factors [18]: Monestrum (which prevents deliberate further pregnancies in most females along the season) and an unusually long diestrous (pseudopregnancy) phase (which puts the non-pregnant, usually young females into a hormonal state that facilitates maternal behaviors). As monestrum and alloparental helper behavior are definitely present in some dogs (dingoes, [27]), as well as pseudopregnancy, which is a pronounced feature even in companion dog females [55], our results (i.e., young females are among the most attentive to whining puppies and respond to puppies in the friendliest manner) are in line with the literature. Breeders also reported that older, intact, and parous dogs showed heightened interest towards whining puppies and were more likely to play with the puppies—this behavior can be expected, according to Schradin et al. [56], who, besides the neoteny-helper hypothesis, highlighted another way of alloparental care: The parent–helper scenario. According to the latter, even adult members of the group can show helping behavior with the puppies, if their endocrine system mimics the changes otherwise typical to the lactating mother.

Finally, breeders reported some level of discrimination by the mother dog regarding her interactions with other dogs in the household. Mothers showed the most agonistic behaviors with other females, if those were either adult, or intact, nulliparous ones. One should take into consideration, of course, that the distribution of sexes and age cohorts were uncontrolled in our sample, which could have an effect on the statistical reliability. On the other hand, there are reports of protective mother dogs (in free-ranging dogs, [57]), with an emphasis on the most frequent agonistic interactions among the adult females [58]. Paul and colleagues [19], however, showed that in free-ranging dogs, grandmothers may provide help with their daughters’ puppies. As in our study, the mother dogs showed higher aggression against intact, nulliparous females, the observations made on companion dogs and on free-ranging dogs (that were accepting help from their mothers) are not excluding each other. However, in general, the reportedly higher occurrence of agonistic behaviors in dog mothers against other (intact) females can be explained as a result of avoiding possible resource competition, and even a probability of infanticide (which is observed among female wolves, [59]).

## 5. Conclusions

Based on the observations of companion dog breeders, our study shows an intricate network of interactions among adult dogs of the household and puppies below the age of weaning. Alloparental behaviors and amicable interactions from the adult dogs dominated the scene, with an eventual stress-reducing effect on the behavior of puppies in the case of alarm barks of the adults. The role of dog–dog interactions during the first two months of life might be an important factor for proper socialization and later problem-free behavior with future canine partners.

## Figures and Tables

**Figure 1 animals-09-01011-f001:**
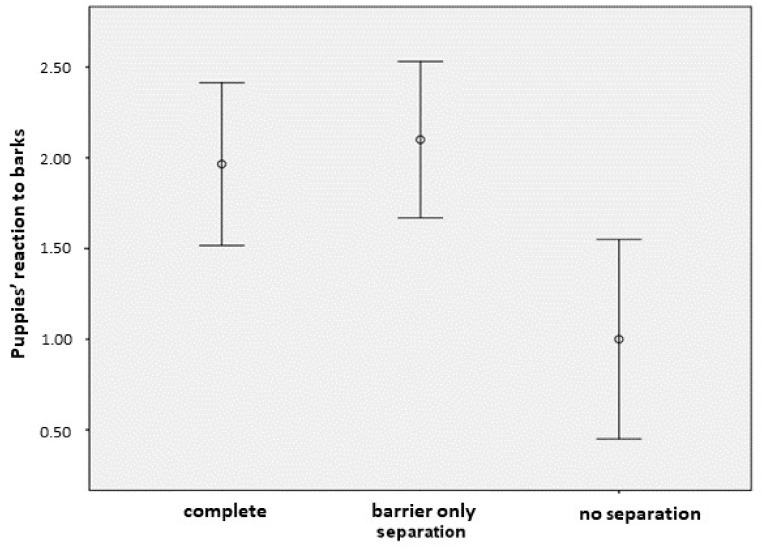
Association between the housing method of the puppies (“separation” 1–3 shows a decreasing level of isolation of the puppies from other dogs) and their reaction to other dogs’ barking (mean ± 95% confidence interval (CI)).

**Figure 2 animals-09-01011-f002:**
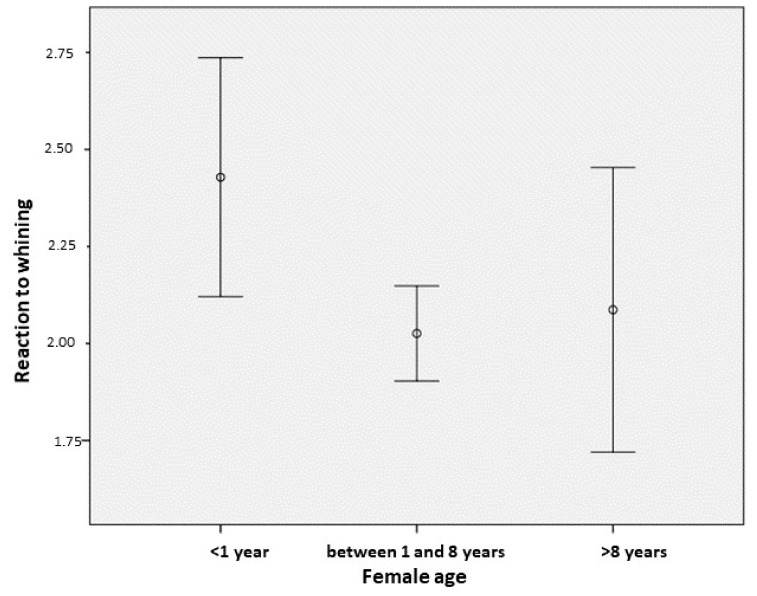
Association between the female dogs’ age (1 = younger than one year; 2 = adult; 3 = older than 8 years) and the intensity of their reaction to the puppies’ whining (mean ± 95% CI).

**Figure 3 animals-09-01011-f003:**
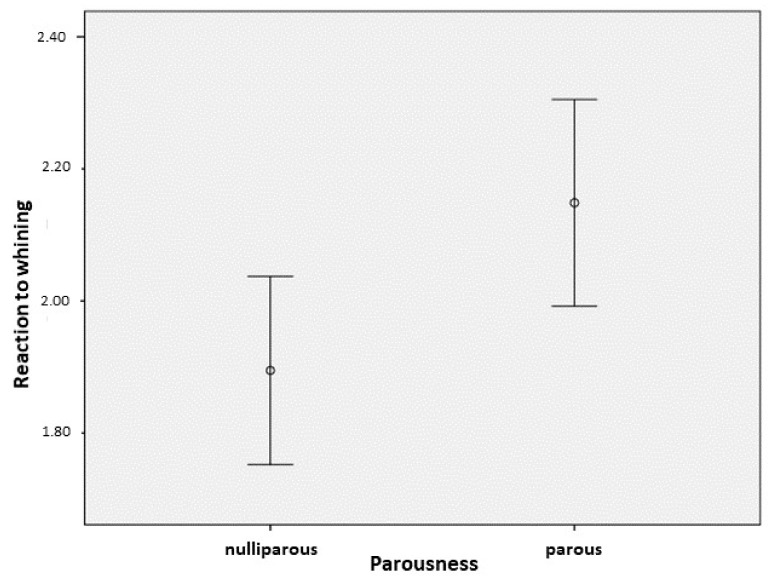
Adult (male + female) dogs’ reaction (mean ± 95% CI) to the whining of puppies in association with the parousness of the adults (0 = did not have puppies; 1 = had puppies before).

**Figure 4 animals-09-01011-f004:**
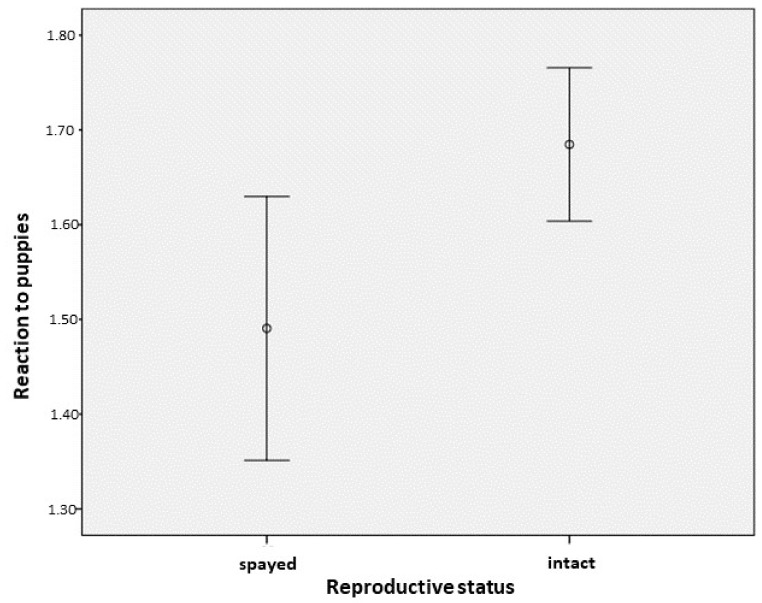
The association between the amicable reaction (mean ± 95% CI) of female dogs to puppies as a function of their reproductive status (0 = spayed, 1 = intact).

**Table 1 animals-09-01011-t001:** Distribution of breeders according to countries and participation in the questionnaire surveys. In the column marked with “Total”, the numbers represent the sum of individual respondents (of each breed) who completed either Questionnaire 1 or 2. FCI = Fédération Cynologique Internationale.

Breed	FCI Breed Group	Total	Questionnaire 2 only	Questionnaires 1 and 2	Country
Australian Shepherd	FCI-1	4	2		Hungary, Germany
Belgian Shepherd Dog/Malinois	FCI-1	2	1	1	Hungary, UK
Border Collie	FCI-1	1	1		Hungary
Bouvier des Flandres	FCI-1	1		1	Hungary
Collie Rough	FCI-1	1	1		Hungary
Collie Smooth	FCI-1	1			Germany
German Shepherd Dog	FCI-1	4			Hungary
Mudi	FCI-1	5	1	1	Hungary
Old English Sheepdog	FCI-1	1			Poland
Pumi	FCI-1	1			Sweden
Shetland Sheepdog	FCI-1	2			Hungary
Cane Corso	FCI-2	1			Hungary
Doberman	FCI-2	4	2	2	Hungary, Germany
Great Dane	FCI-2	2			Hungary
Kangal	FCI-2	1		1	Hungary
Rottweiler	FCI-2	3	1	1	Hungary, US
Schnauzer—Giant	FCI-2	3			Hungary
Schnauzer—Middle	FCI-2	1	1		Hungary
Shar Pei	FCI-2	1			Hungary
Am. Pitbull Terrier	FCI-3	3	1		Hungary
Am. Staffordshire Terrier	FCI-3	2			Hungary
Biewer Yorkshire Terrier	FCI-3	1			Hungary
Jack Russell Terrier	FCI-3	1			Hungary
Yorkshire Terrier	FCI-3	1			Hungary
Dachshund Mini	FCI-4	1			Finland
Dachshund Short Haired	FCI-4	1			Belgium
Akita Inu	FCI-5	1	1		Hungary
German Spitz Klein	FCI-5	2			Hungary, UK
Hokkaido	FCI-5	1			Hungary
Keeshond (Wolfspitz)	FCI-5	1			Germany
Shiba Inu	FCI-5	1			Lithuania
Siberian Husky	FCI-5	3	1		Hungary
Rhodesian Ridgeback	FCI-6	1		1	Germany
Braque d’Auvergne	FCI-7	1			Hungary
Hungarian Vizsla	FCI-7	2		1	Hungary
English Cocker Spaniel	FCI-8	1	1		Hungary
Flatcoated Retriever	FCI-8	1		1	Finland
Labrador Retriever	FCI-8	5	1		Hungary, UK, Italy
Chinese Crested Dog	FCI-9	3		1	Hungary
Bichon Frise	FCI-9	1		1	Hungary
Bolognese	FCI-9	2		1	Hungary, Slovenia
Coton de Tulear	FCI-9	1		1	Hungary
French Bulldog	FCI-9	2		1	Hungary
Havanese	FCI-9	4		1	Hungary
Lhasa Apso	FCI-9	1			Hungary
Poodle—all sizes	FCI-9	3		1	Hungary
Shih Tzu	FCI-9	1			Hungary
Tibetan Terrier	FCI-9	3	1	1	Hungary
Borzoi	FCI-10	1	1		Hungary
Whippet	FCI-10	3		1	Hungary, Austria

**Table 2 animals-09-01011-t002:** Parameter estimates of the puppies’ reaction to other dogs’ barking as a function of the method of housing of the puppies (ordinal regression). “Barkreact” 0–2 levels show an increasing intensity of fearful/joining reaction when other dogs bark. “Separation” 1–3 levels show a decreasing extent of separating the puppies from the other dogs during the time spent at the breeder’s house. * this parameter is set to 0 because it is redundant.

Parameter		Estimate	Std. Error	Wald	df	Sig.	95% Confidence Interval	
							Lower Bound	Upper Bound
Threshold	[barkreact = 0]	−0.317	0.472	0.450	1	0.502	−1.243	0.609
	[barkreact = 1]	0.893	0.486	3.367	1	0.067	−0.061	1.846
	[barkreact = 2]	1.496	0.504	8.809	1	0.003	0.508	2.484
Location	[separation = 1]	1.461	0.589	6.154	1	0.013	0.307	2.615
	[separation = 2]	1.651	0.592	7.771	1	0.005	0.490	2.812
	[separation = 3]	0 *	.	.	0	.	.	.

**Table 3 animals-09-01011-t003:** Parameter estimates of the reactions of adult female dogs to the whining of the puppies as a function of age, reproductive status, and parousness (had offspring or not). GzLM with ordinal logistic. * this parameter is set to 0 because it is redundant.

Parameter		B	Std. Error	95% Wald Confidence Interval		Hypothesis Test		
				Lower	Upper	Wald Chi-Square	df	Sig.
Threshold	[react_whine = 1]	−1.485	0.4545	−2.375	−0.594	10.671	1	0.001
	[react_whine = 2]	0.912	0.4410	0.047	1.776	4.275	1	0.039
[age = below 1 year]		2.113	7824	0.579	3.646	7.292	1	0.007
[age = between 1–8 years]		0.579	0.5375	−0.474	1.633	1.162	1	0.281
[age = older than 8 years]		0 *	.	.	.	.	.	.
[spayed]		−1.152	0.5439	−2.218	−0.086	4.488	1	0.034
[intact]		0	.	.	.	.	.	.
[parous = 0]		−1.140	0.4780	−2.077	−0.203	5.690	1	0.017
[parous = 1]		0 *	.	.	.	.	.	.

**Table 4 animals-09-01011-t004:** Parameter estimates of adult dogs’ reaction to the whining of puppies, as a function of their sex and parousness (0 = did not have puppies; 1 = had puppies). GzLM with ordinal logistic. * this parameter is set to 0 because it is redundant.

Parameter		B	Std. Error	95% Wald Confidence Interval		Hypothesis Test		
				Lower	Upper	Wald Chi-Square	df	Sig.
Threshold	[react_whine = 1]	−2.322	0.3886	−3.083	−1.560	35.694	1	0.000
	[react_whine = 2]	0.396	0.3295	−0.250	1.041	1.441	1	0.230
[male]		−0.963	0.4643	−1.873	−0.053	4.302	1	0.038
[female]		0 *	.	.	.	.	.	.
[parous = 0]		−1.273	0.4984	−2.250	−0.296	6.527	1	0.011
[parous = 1]		0 *	.	.	.	.	.	.

**Table 5 animals-09-01011-t005:** Parameter estimates of the female dogs’ behavior with the puppies in association with their age, reproductive status, and parousness (did not have puppies or had puppies before). GzLM with binary logistics. * this parameter is set to 0 because it is redundant.

Parameter	B	Std. Error	95% Wald Confidence Interval		Hypothesis Test		
			Lower	Upper	Wald Chi-Square	df	Sig.
(Intercept)	0.241	0.4029	−0.549	1.031	0.358	1	0.549
[age = below 1 year]	−2.044	0.7752	−3.564	−0.525	6.955	1	0.008
[age = 1–8 yeas]	−1.259	0.5064	−2.252	−0.266	6.181	1	0.013
[age = older than 8 years]	0 *	.	.	.	.	.	.
[spayed]	0.935	0.3727	0.204	1.666	6.293	1	0.012
[intact]	0	.	.	.	.	.	.
[parous = 0]	0.194	0.3723	−0.536	0.923	0.271	1	0.603
[parous = 1]	0 *	.	.	.	.	.	.

**Table 6 animals-09-01011-t006:** Parameter estimates of the mother dogs’ reaction (agonistic, neutral, friendly) to other females in the household as a function of the other females’ age, reproductive status, and parousness. Significant interaction (*) between reproductive status and parousness is included. GzLM with ordinal logistic. * this parameter is set to 0 because it is redundant.

Parameter		B	Std. Error	95% Wald Confidence Interval		Hypothesis Test		
				Lower	Upper	Wald Chi-Square	df	Sig.
Threshold	[react_female = −1]	−2.019	0.3824	−2.768	−1.269	27.876	1	0.000
	[react_female = 0]	−0.751	0.3556	−1.448	−0.054	4.463	1	0.035
[age = less than 1 year]		1.744	0.6716	0.428	3.061	6.743	1	0.009
[age = 1–8 years]		−0.195	0.3999	−0.979	0.588	0.239	1	0.625
[age = older than 8 years]		0 *	.	.	.	.	.	.
[spayed]		−0.646	0.4022	−1.434	0.142	2.579	1	0.108
[intact]		0 *	.	.	.	.	.	.
[parous = 0]		−2.003	0.4294	−2.844	−1.161	21.748	1	0.000
[parous = 1]		0 *	.	.	.	.	.	.
[reproductive = 0] * [parous = 1]		2.117	0.6065	0.928	3.306	12.183	1	0.000

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
