# Peer review of "Forgotten, But Not Lost—Alloparental Behavior and Pup–Adult Interactions in Companion Dogs"

_animals, 2019, doi:10.3390/ani9121011_

Round 1

Reviewer 1 Report

Title:  Forgotten, but not lost – alloparental behavior and 2 pup-adult interactions in companion dogs

REVIEW ROUND 2

Ethics review

From round 1 of review:

Was there an institutional ethics review of this study? If so, should be stated in the beginning.

RESPONSE: We did not need ethical permission for the study, as the participation to the questionnaire was voluntary, anonymous, and the participants were informed a priori about the goals of the research and the ways and purposes their answers will be stored, analyzed and used.

Reviewer comment: please note in the US, this study might require, at some institutions, a determination from an institutional review board, even if voluntary, anonymous and with informed consent. That said, it would likely be designated exempt from review, which would mean that the protocol would not go to the full committee for review but would still be reviewed by a member of the IRB office (or other review entity). I realize this study was conducted elsewhere and the rules governing it are different, just pointing out one perspective. And now that I have read the manuscript, I see the authors included a review statement & determination so this comment has been addressed.

Method

P5 development of the survey: please add the average time of completion for each survey

P5 line 127: please add the name of the sampling method, as I indicated in the first round of review: “nonprobability convenience sampling”. One way to do this would be: “Nonprobability convenience sampling was used as the questionnaires were distributed via social media, predominantly by sharing them in various Facebook groups, dedicated to the breeding of specific dog breeds, or breeding of purebred dogs in general.”

P5 line 131: suggest changing to: “however, this resulted in only about 10% [or percent per journal standards] of the total sample.”

P5 line 135: change “test” to “survey”.

P5 line 136: delete “any” from the sentence about incentives.

P6 line 143: suggest a minor revision of this sentence: “We requested that the breeders should answer to the questions regarding their observations onof dog-dog interactions based on their ‘entire experience’ of their past litters bred.”

Table 1 – I believe the numbers in the FCI column represent the FCI breed groups? If so, suggest just using those labels as currently it is easy to misconstrue those numbers as sample sizes.

As mentioned previously, more description of the questionnaire is needed, particularly around the response options. It would be helpful to include a brief description of the response options participants could choose from, so readers know what options participants saw & choose from. Or if there were no options given and everything was open-ended. Or if the questions were single-select or multiple-select. The questionnaire is the instrument for measuring the survey constructs and therefore the results are only as good as the instrument used. Knowing more about the questions & response options would enable readers to make a judgment of how the questions might affect the results. Providing the link to the survey on Google Form is extremely helpful but the authors can’t rely on that to fully inform their readers of such an important aspect of their study; a bit more description is needed in the text of the article.

Results

Figures: while the interpretability of the figures has improved with the revisions made, they are still difficult to understand without careful reading of the figure note or method. Best practice generally advice tables & figures “stand on their own”. I believe it would be easy to add more detail to the axis labels and/or units of measurement to make the tables easier to interpret. For example, in Figure 1, what does the #1 represent for the variable/concept “separation”? Even if the authors used descriptive terms like “Significant isolation (1)” “Moderate isolation (2)” “Mild isolation (3)”, it would help the reader (these might need to be defined in variables or data analysis). What does a 2.5 represent for the concept “puppies’ reaction to barks”? Even if it were difficult to label the actual measurement units, a brief mention in the title would help. Figure 3 and 4 have numbered categories on the X axis that could easily be replaced with the label (had puppies, did not have puppies) that would help tremendously.

Discussion

Nice addition with the new section about limitations. One other that could be mentioned is the geographic focus.

General/Minor Comments

--I don’t believe there’s a need for “e.g.,” when immediately including a citation, as that is what the citation is for. For example, P3 line 62: “biology of dogs (e.g., [21]), as free ranging dogs…” can be changed to “biology of dogs [21], as free ranging dogs…”.

--Similarly to e.g. need a comma after, same deal for i.e. -- i.e.,

--P5 line 114, P17 line 322: e.g. needs a comma

--P8 line 152 “puppies” is plural possessive so should be “puppies’” (apostrophe at end).

--P8 line 153 paragraph should be indented

Author Response

Dear Reviewer 1,

Thank you again for the positive comments and suggestions regarding our manuscript. Please find the detailed list of our responses below.

Method

P5 development of the survey: please add the average time of completion for each survey

RESPONSE: We added the requested detail (Lines 124-125) – sorry for that we forgot it previously.

 P5 line 127: please add the name of the sampling method, as I indicated in the first round of review: “nonprobability convenience sampling”. One way to do this would be: “Nonprobability convenience sampling was used as the questionnaires were distributed via social media, predominantly by sharing them in various Facebook groups, dedicated to the breeding of specific dog breeds, or breeding of purebred dogs in general.”

RESPONSE: Thank you for the specific suggestion, we incorporated it to the manuscript (Lines 128-130).

P5 line 131: suggest changing to: “however, this resulted in only about 10% [or percent per journal standards] of the total sample.”

 RESPONSE: Thank you, done.

P5 line 135: change “test” to “survey”.

RESPONSE: Thank you, done.

P5 line 136: delete “any” from the sentence about incentives.

RESPONSE: Thank you, done.

P6 line 143: suggest a minor revision of this sentence: “We requested that the breeders shouldanswer to the questions regarding their observations onof dog-dog interactions based on their ‘entire experience’ of their past litters bred.”

RESPONSE: Thank you for the specific suggestion, we incorporated it to the manuscript (Lines 145-146).

Table 1 – I believe the numbers in the FCI column represent the FCI breed groups? If so, suggest just using those labels as currently it is easy to misconstrue those numbers as sample sizes.

RESPONSE: We changed the FCI-group labels to a less ambiguous version in Table 1.

As mentioned previously, more description of the questionnaire is needed, particularly around the response options. It would be helpful to include a brief description of the response options participants could choose from, so readers know what options participants saw & choose from. Or if there were no options given and everything was open-ended. Or if the questions were single-select or multiple-select. The questionnaire is the instrument for measuring the survey constructs and therefore the results are only as good as the instrument used. Knowing more about the questions & response options would enable readers to make a judgment of how the questions might affect the results. Providing the link to the survey on Google Form is extremely helpful but the authors can’t rely on that to fully inform their readers of such an important aspect of their study; a bit more description is needed in the text of the article.

RESPONSE: We indicated in the revision whether the questions were single or multiple choice items.

 Results

Figures: while the interpretability of the figures has improved with the revisions made, they are still difficult to understand without careful reading of the figure note or method. Best practice generally advice tables & figures “stand on their own”. I believe it would be easy to add more detail to the axis labels and/or units of measurement to make the tables easier to interpret. For example, in Figure 1, what does the #1 represent for the variable/concept “separation”? Even if the authors used descriptive terms like “Significant isolation (1)” “Moderate isolation (2)” “Mild isolation (3)”, it would help the reader (these might need to be defined in variables or data analysis). What does a 2.5 represent for the concept “puppies’ reaction to barks”? Even if it were difficult to label the actual measurement units, a brief mention in the title would help. Figure 3 and 4 have numbered categories on the X axis that could easily be replaced with the label (had puppies, did not have puppies) that would help tremendously.

RESPONSE: We added more informative labels to the X-axis on each figure.

Discussion

Nice addition with the new section about limitations. One other that could be mentioned is the geographic focus.

RESPONSE: We added this detail (Lines 354-355) to the Discussion. However, we have to mention that reaching a representative multi-national sample, a questionnaire translated to many languages would be needed (plus reliable distribution on various national internet forums), which is definitely a staggering task.

General/Minor Comments

--I don’t believe there’s a need for “e.g.,” when immediately including a citation, as that is what the citation is for. For example, P3 line 62: “biology of dogs (e.g., [21]), as free ranging dogs…” can be changed to “biology of dogs [21], as free ranging dogs…”.

 --Similarly to e.g. need a comma after, same deal for i.e. -- i.e.,

 --P5 line 114, P17 line 322: e.g. needs a comma

 --P8 line 152 “puppies” is plural possessive so should be “puppies’” (apostrophe at end).

 --P8 line 153 paragraph should be indented

RESPONSE: Corrections were done.

Reviewer 2 Report

Well done, just a few more minor comments on the revised draft for english correction. 

Line 136 - change "No any form of incentive was ..." -> "No incentive was..."

Line 159 - flagging this for grammar check, i have never seen "; e.g.," before and recommend rewriting as it just "looks" wrong, sorry no more detail

Line 273 - should actually state "mean +- 95%CI" - remember that the reader should be able to interpret the statistical meaning of the graph in isolation of the remainder of paper. This is not a personal vendetta of mine I promise! I am just trying to help. It was drilled into me during my PhD as a part of my statistical course that all scientific journal article figures should have this information present for proper interpretation. Technically, you are also supposed to have the "n" for each of the 3 groups present on the figure too (if you ask an Australian statistician). You could consider including the "n" in the figure, but I don't mind personally as you can find it in other places within the paper and it is used in the calculation of the CI so is partially contained in the data presented.

Line 284 - Again, include "mean" (or "average" if you prefer, just be consistent throughout the paper)

399 - This sentence is too long, which also makes it difficult to follow. Remove "understandably," and instead put a period "." then start the new sentence with "Most studies have focused on dog-human inter..." (rather than "most efforts were aimed at dog-human inter...")

411 - there is a closing bracket that shouldn't be there "([52]" -> "[52]"

You do not need to make comments on these changes for me, I would be happy with the article once they are made. Congratulations on the publication!

Kind regards,

Dennis

Author Response

Dear Dennis (aka Reviewer 2),

Thank you very much for the support and helpful suggestions. Please find our detailed answers below.

REVIEWER 2

Well done, just a few more minor comments on the revised draft for english correction. 

Line 136 - change "No any form of incentive was ..." -> "No incentive was..."

RESPONSE: Corrected, thank you.

Line 159 - flagging this for grammar check, i have never seen "; e.g.," before and recommend rewriting as it just "looks" wrong, sorry no more detail

RESPONSE: Between Reviewer 1 and 2, I am getting confused with the correct way of writing the e.g. I hope that the copyeditor of Animals can provide the solution.

Line 273 - should actually state "mean +- 95%CI" - remember that the reader should be able to interpret the statistical meaning of the graph in isolation of the remainder of paper. This is not a personal vendetta of mine I promise! I am just trying to help. It was drilled into me during my PhD as a part of my statistical course that all scientific journal article figures should have this information present for proper interpretation. Technically, you are also supposed to have the "n" for each of the 3 groups present on the figure too (if you ask an Australian statistician). You could consider including the "n" in the figure, but I don't mind personally as you can find it in other places within the paper and it is used in the calculation of the CI so is partially contained in the data presented.

Line 284 - Again, include "mean" (or "average" if you prefer, just be consistent throughout the paper)

RESPONSE: “mean” was added to each figure captions.

399 - This sentence is too long, which also makes it difficult to follow. Remove "understandably," and instead put a period "." then start the new sentence with "Most studies have focused on dog-human inter..." (rather than "most efforts were aimed at dog-human inter...")

RESPONSE: Thank you for the helpful comment, we rewrote the text accordingly.

411 - there is a closing bracket that shouldn't be there "([52]" -> "[52]"

You do not need to make comments on these changes for me, I would be happy with the article once they are made. Congratulations on the publication!

Kind regards,

Dennis

RESPONSE: Thank you very much!

This manuscript is a resubmission of an earlier submission. The following is a list of the peer review reports and author responses from that submission.

Round 1

Reviewer 1 Report

.

Author Response

Response letter for the revised manuscript

“Forgotten, but not lost – alloparental behavior and pup-adult interactions in companion dogs”

written by Pongrácz & Sztruhala

Dear Editors and Reviewers,

We were very pleased when read the positive and supportive comments and opinions about our manuscript. We found the Reviewers’ suggestions and questions to be very useful, and we reworked the text accordingly. Please find our detailed answers below, as well as the revised manuscript with the highlighted changes throughout.

We hope that the new version of the manuscript will better fit to the standards of the journal Animals now.

Sincerely,

Péter Pongrácz, PhD

(corresponding author)

RESPONSE TO REVIEWER 1

Thank you for the positive evaluation!

As there was no detailed review, we cannot provide more details to this Reviewer.

Reviewer 2 Report

Dear Authors,

This was a high-quality manuscript, and it presents some useful and novel information. Overall, the English and structure was excellent, very readable. Most of my comments are minor clarifications or changes that should be easily completed. While the behavioural and evolutionary ethology is clearly your interest and focus, I am pleased you have recognised the relevance of this work to modern-day issues with dog behaviour and aggression.  

It was a pleasure to read and comment on this work. Good luck with your other reviews. Please find my comments below.

Dennis Wormald

27 - "We made an international internet survey, asking companion dog breeders about the interactions" -> "An international online survey of companion dog breeders was conducted, asking about the interactions..."

133 - Table 1 - I am assuming that total includes breeders who only completed questionnaire 1. However this is not clear to me, please clarify somewhere in the manuscript.

133 - Table 1 - As a suggestion, a summary table might be better, using breed groups categorised according to the FCI system, particularly since you used this for your analysis

190 - If alloparental behaviour was a binary variable, how were the two binary groups defined? I assume it was based on the presence/absence of ANY alloparental behaviour? Please clarify here.

191 - I assume gzlm is generalised linear model? Please clarify in manuscript.

192 - Again, do you mean the presence/absence of regurgitating behaviour as your binary variable?

201-201 - Please have the journal editor check this section for style of headings/subheading etc. as I am unfamiliar with the journal format.

209 - I don't see the point of this statistical test, why do we need to compare the observed frequency of regurgitation to anything at all? It only needs to be reported in my opinion. I think it is a very interesting report.

215 - The same as comment for 209 - there is no need for statistical testing of this particular data, please just report the observed frequencies. If you want to do statistics, you could do a binomial 95% confidence interval on the population proportion based on this sample.

219 - Table 2 - I am unsure of the format of this journal, in all the international english papers I read, I am used to a period (.) being used rather than a commar (,) to denote a decimal point.

242 - Figure 2 - the 95% CI should be written in the table legend (e.g. values represent mean reaction to whinin +- 95% CI), not on the graph axis, the y axis should be labelled something like "reaction to whining". The table legend should also state the corresponding number for each group like "(1-younger than one year, 2-adult, 3-older than 8 years)"

255 - Figure 3 - the 95% CI should be written in the table legend, not on the graph axis, the y - axis should be labelled something like "reaction to whining"

260 - Figure 4 and legend have the same problems as figures 2 and 3 - please fix

299 - "dogs and, parous" - please fix

303 - This sentence seems out of place, I thought it was not logically connected to the following sentence or the paragraph. Then I realised that if you write "compared to wild dogs" at the end of the sentence, it all makes sense. 

305 - "( see a review [43])" -> "(see review [43])"

308 - Now I understand what the paragraph is about, I feel this last sentence could go first and it would be much clearer to me. 

336 - "procurement), in companion" -> "procurement). In companion"

338 - "effect either – of course, we have to remember that the sample size was low" -> "effect, however this could be explained by the low sample size of individual breed groups"

345 - As I am sure you know, there is little published material on this, particularly recent material. I would however recommend that you read the Fisher dissertation if you haven't yet. "Fisher, A.E., 1955. The Effects of Differential Early Treatment on the Social and Exploratory Behavior of Puppies, Doctoral Dissertation, Pennsylvania State University, University Park Campus." - From page 45 there are results on social deprivation of early dog-dog interactions on puppies. You might want to consider citing this in your discussion if you feel appropriate.

351 - 355 - In my own Australian study (cited below), I found that the earlier that a dog owner exposes their puppy to other dogs in public to unknown dogs, the more likely the dog was to actually end up with dog-dog aggression at an older age. This ties in excellently with your present study. Dog-dog interactions with unknown dogs in public are not likely to include alloparental behaviours. The risk of negative experiences during this early sensitive age is high when owners socialise their dog incorrectly. In contrast, alloparental behaviours would be exptected to be positive and not fearful, providing good social experiences. Together, our two studies show the contrast in effectiveness of different types of social exposure in the puppy at preventing fear aggression. I believe we are showing that a puppy should be kept at home with the breeder and multiple other familiar dogs and then continue to socialise with familiar dogs throughout the early socialisation period. A long history of positive interactions with other dogs is likely to be protective against developing fear when the puppy is finally socialised with unfamiliar dogs. 

Wormald, Dennis, et al. "Analysis of correlations between early social exposure and reported aggression in the dog." Journal of Veterinary Behavior 15 (2016): 31-36.

351 - 355 - In the reference here to the results from line 220, It should also be noted that correlation is not causation. It is possible that puppies that were separated from other dogs, were separated because the other dogs were aggressive. The breeder may have done this to protect the puppies. In this case they may have been frightened by the aggressive dog, or attacked and then separated. Similarly the puppy could be related to other dogs that are aggressive and therefore be separated for safety and also carry the aggressive genetic or epigenetic traits.

I was happy with the remainder of the manuscript.

Author Response

Response letter for the revised manuscript

“Forgotten, but not lost – alloparental behavior and pup-adult interactions in companion dogs”

written by Pongrácz & Sztruhala

Dear Editors and Reviewers,

We were very pleased when read the positive and supportive comments and opinions about our manuscript. We found the Reviewers’ suggestions and questions to be very useful, and we reworked the text accordingly. Please find our detailed answers below, as well as the revised manuscript with the highlighted changes throughout.

We hope that the new version of the manuscript will better fit to the standards of the journal Animals now.

Sincerely,

Péter Pongrácz, PhD

(corresponding author)

RESPONSES TO REVIEWER 2

Thank you for the very supportive and helpful comments!

27 - "We made an international internet survey, asking companion dog breeders about the interactions" -> "An international online survey of companion dog breeders was conducted, asking about the interactions..."

RESPONSE: Thank you for the correction – we used the suggested version of wording in our revision (Lines 28-29).

133 - Table 1 - I am assuming that total includes breeders who only completed questionnaire 1. However this is not clear to me, please clarify somewhere in the manuscript.

RESPONSE: We made it more clear now in the caption of Table 1, what the numbers mean in each column.

133 - Table 1 - As a suggestion, a summary table might be better, using breed groups categorised according to the FCI system, particularly since you used this for your analysis

RESPONSE: According to the suggestion, we reorganized Table 1, now the dogs are listed according to the FCI breed groups.

190 - If alloparental behaviour was a binary variable, how were the two binary groups defined? I assume it was based on the presence/absence of ANY alloparental behaviour? Please clarify here.

RESPONSE: We added details (Line 228).

191 - I assume gzlm is generalised linear model? Please clarify in manuscript.

RESPONSE: We added the full name of the method here (Line 229).

192 - Again, do you mean the presence/absence of regurgitating behaviour as your binary variable?

RESPONSE: We added details (Line 232).

201-201 - Please have the journal editor check this section for style of headings/subheading etc. as I am unfamiliar with the journal format.

RESPONSE: We tried to mark different levels of sub-chapters with clearly identifiable subheadings, however, we leave the final format on the Editor’s decision.

209 - I don't see the point of this statistical test, why do we need to compare the observed frequency of regurgitation to anything at all? It only needs to be reported in my opinion. I think it is a very interesting report.

RESPONSE: Thank you for the encouraging comment. We believe that by running the mentioned statistical tests, it helps to place our results into a perspective of different aspects that feeding by regurgitation can be expected in various populations of dogs (according to the literature).

215 - The same as comment for 209 - there is no need for statistical testing of this particular data, please just report the observed frequencies. If you want to do statistics, you could do a binomial 95% confidence interval on the population proportion based on this sample.

RESPONSE: See our comment above. We think that the statistical test does not weaken the result – however, if the Editor decides that the statistical analysis here is superfluous, we can resort to reporting the frequencies only.

219 - Table 2 - I am unsure of the format of this journal, in all the international english papers I read, I am used to a period (.) being used rather than a comma (,) to denote a decimal point.

RESPONSE: Thank you for the note, we corrected each table accordingly.

242 - Figure 2 - the 95% CI should be written in the table legend (e.g. values represent mean reaction to whinin +- 95% CI), not on the graph axis, the y axis should be labelled something like "reaction to whining". The table legend should also state the corresponding number for each group like "(1-younger than one year, 2-adult, 3-older than 8 years)"

255 - Figure 3 - the 95% CI should be written in the table legend, not on the graph axis, the y - axis should be labelled something like "reaction to whining"

260 - Figure 4 and legend have the same problems as figures 2 and 3 - please fix

RESPONSE: We revised the figures and figure legends according to this request.

299 - "dogs and, parous" - please fix

RESPONSE: Fixed, thank you.

303 - This sentence seems out of place, I thought it was not logically connected to the following sentence or the paragraph. Then I realised that if you write "compared to wild dogs" at the end of the sentence, it all makes sense. 

RESPONSE: Thank you for the note, we corrected the text as suggested.

305 - "( see a review [43])" -> "(see review [43])"

RESPONSE: Fixed, thank you.

308 - Now I understand what the paragraph is about, I feel this last sentence could go first and it would be much clearer to me. 

RESPONSE: Thank you for the note, we corrected the text as suggested (Lines 381-383).

336 - "procurement), in companion" -> "procurement). In companion"

RESPONSE: Fixed, thank you.

338 - "effect either – of course, we have to remember that the sample size was low" -> "effect, however this could be explained by the low sample size of individual breed groups"

RESPONSE: Thank you for the note, we corrected the text as suggested

345 - As I am sure you know, there is little published material on this, particularly recent material. I would however recommend that you read the Fisher dissertation if you haven't yet. "Fisher, A.E., 1955. The Effects of Differential Early Treatment on the Social and Exploratory Behavior of Puppies, Doctoral Dissertation, Pennsylvania State University, University Park Campus." - From page 45 there are results on social deprivation of early dog-dog interactions on puppies. You might want to consider citing this in your discussion if you feel appropriate.

RESPONSE: Thank you for the very useful suggestion, we added this reference to the discussion.

351 - 355 - In my own Australian study (cited below), I found that the earlier that a dog owner exposes their puppy to other dogs in public to unknown dogs, the more likely the dog was to actually end up with dog-dog aggression at an older age. This ties in excellently with your present study. Dog-dog interactions with unknown dogs in public are not likely to include alloparental behaviours. The risk of negative experiences during this early sensitive age is high when owners socialise their dog incorrectly. In contrast, alloparental behaviours would be exptected to be positive and not fearful, providing good social experiences. Together, our two studies show the contrast in effectiveness of different types of social exposure in the puppy at preventing fear aggression. I believe we are showing that a puppy should be kept at home with the breeder and multiple other familiar dogs and then continue to socialise with familiar dogs throughout the early socialisation period. A long history of positive interactions with other dogs is likely to be protective against developing fear when the puppy is finally socialised with unfamiliar dogs. 

Wormald, Dennis, et al. "Analysis of correlations between early social exposure and reported aggression in the dog." Journal of Veterinary Behavior 15 (2016): 31-36.

RESPONSE: Thank you for this excellent reference and the suggested connection with our results – we totally agree, therefore incorporated this thought and citation to the manuscript.

351 - 355 - In the reference here to the results from line 220, It should also be noted that correlation is not causation. It is possible that puppies that were separated from other dogs, were separated because the other dogs were aggressive. The breeder may have done this to protect the puppies. In this case they may have been frightened by the aggressive dog, or attacked and then separated. Similarly the puppy could be related to other dogs that are aggressive and therefore be separated for safety and also carry the aggressive genetic or epigenetic traits.

RESPONSE: We agree with this note – we avoided to use the term “effect”, or “affects” regarding the mentioned result about the social aspects of housing conditions of the puppies and their subsequent reactions to other dogs’ barking. We carefully used the term “association” between the factor and the dependent variable, acknowledging the correlative nature of this result, where those circumstances that were mentioned by the Reviewer can easily affect the behavioral epigeny of the puppies.

Reviewer 3 Report

Title:  Forgotten, but not lost – alloparental behavior and 2 pup-adult interactions in companion dogs

Summary: A well-written and articulate article that reports the results of an internet survey of a convenience sample of breeders reporting on their dogs’ alloparenting behavior.

General Comments:

The justification for this study is articulated well and very clear. Discussion reads well and nicely situates the findings within the field. More detail is needed on the method on the sampling method (relates directly to important limitations not discussed) and measures (more information needed on the response options to questions, especially since past research has demonstrated that people, even owners, are not always that knowledgeable about dog behavior and that how the question is asked can influence the responses). Related, more discussion of the study’s limitations is needed, especially as related to sample size and sampling method (and their effects on generalizability) and question construction (and effects on validity).

Specific Comments:

Simple Summary

P1 line 9: replace “few” with “little”: however we know surprisingly few about their natural parental behaviors. P1 line 10: add “although”: “Meanwhile, although wolves…”

Abstract

P1 line 27: replace “made” with “conducted”

Keywords

Since you have room for one more, you may want to add “behavior”

Intro

P2 line 62: the line that starts “many authors” could use citations to support this statement. P2 line 91: “group member”, do you mean household member? Vs pack which may or may not be contained within a household? P3 line 96: suggest changing “applied” to “conducted”

Method

Was there an institutional ethics review of this study? If so, should be stated in the beginning. The beginning of the section “Development of the questionnaire” appears to be less about the actual development of the questionnaire and more an explanation of early puppy development, perhaps as a rationale for why the questionnaire focused on this period, which may fit better under variables? Or make this section just about the questionnaire and add in all the variable information so there is more information about the questionnaire/questions? It may be useful to add a section “procedure” and move all details related to the administration of the survey there. Such as the sampling plan, survey hosting, data collection dates, time to complete, etc. Then the subjects section would be just about the eligibility requirements and who participated in the survey. More detail about the sampling plan and administration of the survey would be helpful. Such as what platform hosted the survey (I assume Google Drive), how long did it take for participants to complete it, was an incentive offered. The actual sampling method is non-probability convenience sampling (if snowball sampling was also used, add that). What social media platforms were used and were only personal networks used or were other networks leveraged on these platforms? For example, the researchers could have posted to their own personal networks on Facebook, which may have resulted in a skewed sample (or not), or they could have posted to breeder groups on Facebook, which may be unassociated with the researchers. Or they could have boosted posts or used advertising. Email was also used to distribute the survey – to whom? Who emailed the invitation and to whom? How were these people selected? Since there appears to have been two survey efforts, this information should be included for both surveys unless they used exactly the same sources and sampling methods. Was the questionnaire anonymous or confidential? Were there any eligibility requirements as to who actually responded to the questionnaire? Was it anyone in the household (including children?) or was there any screening as to the person who makes the breeding decisions, runs the breeding program/business, knows the dogs best, etc.? While Table 1 presents the distribution of participants by breed and country, I would be satisfied to read in text the number of participants by country and maybe the top 5 breeds. I leave that to your, and editor, discretion. P5 – how were breeders instructed to respond when they answered for their dogs – one pup (from when), the last litter, the last year, all litters ever? P5 – it would be helpful to know the answer options for some of these variables, for example, was alloparental behavior (nursing) asked with yes/no/don’t know response options or with frequency response options (never, one time, several times, etc.)? (from the information presented in the data analysis section, it appears to be a yes/no question but it would be useful to know that head of time) Was it just one question or was it a check all that apply format with a variety of alloparental behaviors listed? While readers could visit the questionnaire links you helpfully provided, for those not so inclined, more description of not only the questions but the response options would be very helpful. P5 – for single-select questions that asked about the behavior of multiple individuals, how were breeders instructed to respond if different individuals did different things? For example, puppies’ reaction to adult dogs barking – what if some puppies reacted fearfully but some did not? This relates back to the question above about ‘scope’ – for whom were breeders instruction to respond – an individual dog, a litter (from when), their entire experience? P6 line 170 – it would helpful to mention that paternal status of other dog(s) was tracked as that comes up later in the analysis. P6 line 194 – it is not clear to me why you chose to use ordinal regression for the behavioral reactions when there doesn’t appear to be a clear rank order to the behaviors (ex., puppies have no reaction, puppies run away/are quiet (fear), puppies bark (which could be assertive/aggressive response, fear response, playful, etc.,). It would be helpful to specify what assumptions were checked for the tests (e.g., test of parallel lines for ordinal logistic regression) and how the models fared in them. Alpha of .05 was used to determine significance, I assume?

Results

In general, it would help the reader if the dependent variables were reported in terms of their actual measurements in the results. A statement like “females dogs under one year of age showed the strongest reaction” to puppies’ whining is difficult to interpret without referring back to the method. A small clarification of what the actual reaction was would make this result more impactful. Figures would be easier to interpret with this information as well (rather than just numbers 1, 2, 3). P6 line 212 – is Pal 2015 a citation? In that case, it needs to be replaced with a citation number.

Discussion

P13 line 312 – this conclusion seems to be rather far-reaching given the small sample size and limited number of countries and breeds included. Well-written discussion which nicely situates this study’s findings in the larger field. While there is some mention of limitations, however, more discussion of specific limitations, and their impact on generalization of results, is warranted. Specifically, although the small sample was addressed (but could use more discussion), the sampling method was not. Given the likely biases that may have resulted from an online convenience sample, especially sourcing from social media, a discussion of the limited number of countries and the possible effect of a non-response bias (what about breeders without (reliable?) internet, or those not on social media, or those that didn’t want to respond due to problematic behaviors in their litters or ‘packs’?) is needed. Additionally, there is just one brief mention of breeders’ interpretation of dog behavior. Given the large number of studies examining people’s/owner’s lack of knowledge or misperceptions about dog behavior, there should be a more extensive discussion of this limitation. This survey is entirely self-report which, while capitalizing on a strength, is also a limitation, particularly related to validity. This also relates to the limited information about the questions and response options mentioned above; there is some research examining how different methods of asking questions about dog behavior can result in different answers; therefore, it is important that the response options be described and then discussed in the discussion.

Minor Grammatical Comments:

In US English, “e.g.” is always followed by a comma: “e.g.,” Line 61, “millions” should be singular in this case according to common US usage Line 76, “year” should be plural; and suggest changing “sustain themselves mainly from…” to “by” Line 78, need a closing parenthesis Line 84, comma at the end of the sentence after “environment” is not needed Line 90, need a closing parenthesis (here and elsewhere)

Author Response

Response letter for the revised manuscript

“Forgotten, but not lost – alloparental behavior and pup-adult interactions in companion dogs”

written by Pongrácz & Sztruhala

Dear Editors and Reviewers,

We were very pleased when read the positive and supportive comments and opinions about our manuscript. We found the Reviewers’ suggestions and questions to be very useful, and we reworked the text accordingly. Please find our detailed answers below, as well as the revised manuscript with the highlighted changes throughout.

We hope that the new version of the manuscript will better fit to the standards of the journal Animals now.

Sincerely,

Péter Pongrácz, PhD

(corresponding author)

RESPONSES TO REVIEWER 3

We are grateful for the useful and supportive comments and criticism.

The justification for this study is articulated well and very clear. Discussion reads well and nicely situates the findings within the field. More detail is needed on the method on the sampling method (relates directly to important limitations not discussed) and measures (more information needed on the response options to questions, especially since past research has demonstrated that people, even owners, are not always that knowledgeable about dog behavior and that how the question is asked can influence the responses). Related, more discussion of the study’s limitations is needed, especially as related to sample size and sampling method (and their effects on generalizability) and question construction (and effects on validity).

RESPONSE: We provide a detailed set of answers regarding the development of the survey, and we also added a paragraph to the Discussion where we muster the potential limitations of the study.

Simple Summary

P1 line 9: replace “few” with “little”: however we know surprisingly few about their natural parental behaviors. P1 line 10: add “although”: “Meanwhile, although wolves…”

RESPONSE: Fixed, thank you.

Abstract

P1 line 27: replace “made” with “conducted”

RESPONSE: Fixed, thank you.

Keywords

Since you have room for one more, you may want to add “behavior”

RESPONSE: Fixed, thank you.

Intro

P2 line 62: the line that starts “many authors” could use citations to support this statement.

RESPONSE: Thank you for the suggestion, we modified this sentence thus now it includes the reference.

P2 line 91: “group member”, do you mean household member? Vs pack which may or may not be contained within a household?

RESPONSE: We changed the sentence to a clearer wording (originally, we meant here “other adult household dogs”).

P3 line 96: suggest changing “applied” to “conducted”

Method

Was there an institutional ethics review of this study? If so, should be stated in the beginning.

RESPONSE: We did not need ethical permission for the study, as the participation to the questionnaire was voluntary, anonymous, and the participants were informed a priori about the goals of the research and the ways and purposes their answers will be stored, analyzed and used.

The beginning of the section “Development of the questionnaire” appears to be less about the actual development of the questionnaire and more an explanation of early puppy development, perhaps as a rationale for why the questionnaire focused on this period, which may fit better under variables? Or make this section just about the questionnaire and add in all the variable information so there is more information about the questionnaire/questions? It may be useful to add a section “procedure” and move all details related to the administration of the survey there. Such as the sampling plan, survey hosting, data collection dates, time to complete, etc. Then the subjects section would be just about the eligibility requirements and who participated in the survey.

RESPONSE: Thank you for this suggestion. We revised the Methods section following the Reviewer’s request (Lines 134-188). Now there is a separate subchapter that handles the details of developing the survey, which is followed by the section that explains who could participate in the survey. Finally in another subchapter we give the detailed description of the variables.

More detail about the sampling plan and administration of the survey would be helpful. Such as what platform hosted the survey (I assume Google Drive),

RESPONSE: We agree with the Reviewer that more details is needed about the methods. Beside the detailed answers provided here to these questions, we also incorporated most of these information now to the text (Lines ???). We used Google Forms: Free Online Surveys for Personal Use.

how long did it take for participants to complete it, was an incentive offered.

RESPONSE: Completing questionnaire 1 took 10-15 min (depends on the categories of different adult dogs owned by the particular breeder. There was no any form of incentive offered.

The actual sampling method is non-probability convenience sampling (if snowball sampling was also used, add that). What social media platforms were used and were only personal networks used or were other networks leveraged on these platforms? For example, the researchers could have posted to their own personal networks on Facebook, which may have resulted in a skewed sample (or not), or they could have posted to breeder groups on Facebook, which may be unassociated with the researchers. Or they could have boosted posts or used advertising. Email was also used to distribute the survey – to whom? Who emailed the invitation and to whom? How were these people selected? Since there appears to have been two survey efforts, this information should be included for both surveys unless they used exactly the same sources and sampling methods.

RESPONSE: We did not use ‘snowball sampling’ and advertising for the survey. We followed the same method of distribution in case of both questionnaires. One of the authors (Sára Sztruhala) distributed the survey via social media (FB), where the Hungarian and the English versions were posted to 10-15 various Facebook groups, comprising dog breeders and enthusiasts. Among these FB groups there were both breed-specific and all-breed types. During the sampling period, re-sharing of the survey’s link was repeated in a weekly manner. Besides this, the survey was shared by both authors with a limited number of known breeder acquaintances via email (this did not take more than approximately the 10% of the final N of the sample).

Was the questionnaire anonymous or confidential? Were there any eligibility requirements as to who actually responded to the questionnaire? Was it anyone in the household (including children?) or was there any screening as to the person who makes the breeding decisions, runs the breeding program/business, knows the dogs best, etc.?

RESPONSE: Providing the name of the responder or any contact address (including the email address) was not mandatory for the participation, however, participants provided at least some sort of identifier (name or email) in about 90% of the sample. We requested that the actual responder (who completes the questionnaire) should be always the breeder him/herself.

While Table 1 presents the distribution of participants by breed and country, I would be satisfied to read in text the number of participants by country and maybe the top 5 breeds. I leave that to your, and editor, discretion.

RESPONSE: Table 1 was reorganized upon the request of another Reviewer. In its present form it shows the participants in a grouping based on the FCI dog breed groups.

P5 – how were breeders instructed to respond when they answered for their dogs – one pup (from when), the last litter, the last year, all litters ever?

RESPONSE: We requested the answers to be based on the experience of the given breeder regarding ‘all litters ever’ bred.

P5 – it would be helpful to know the answer options for some of these variables, for example, was alloparental behavior (nursing) asked with yes/no/don’t know response options or with frequency response options (never, one time, several times, etc.)? (from the information presented in the data analysis section, it appears to be a yes/no question but it would be useful to know that head of time) Was it just one question or was it a check all that apply format with a variety of alloparental behaviors listed? While readers could visit the questionnaire links you helpfully provided, for those not so inclined, more description of not only the questions but the response options would be very helpful.

RESPONSE: In the final analysis, both alloparental behaviors and feeding by regurgitation was treated as a binary (presence/absence) variable. However, in the original questionnaires the participants could choose from multiple options (such as the relationship between the mother dog and the other female that nursed the puppies: another daughter, grandmother, unrelated female etc.). Due to the relatively small sample size and the various answers, we decided to group all the responses under the outcome ‘alloparental behavior is present’, where the participant marked at least one type of the listed options as it has been observed by him/her. Besides allo-nursing, other forms of alloparenting were also offered as options in the questionnaires (cleaning, play, regurgitation etc.).

P5 – for single-select questions that asked about the behavior of multiple individuals, how were breeders instructed to respond if different individuals did different things? For example, puppies’ reaction to adult dogs barking – what if some puppies reacted fearfully but some did not? This relates back to the question above about ‘scope’ – for whom were breeders instruction to respond – an individual dog, a litter (from when), their entire experience?

RESPONSE: At the beginning of the questionnaires we instructed the participants that they should base their answers on their entire experience about dogs that they bred so far. At the same time, in case of the items mentioned by the Reviewer, we provided an ‘other’ answer option, in case the breeder wanted to elaborate the response. In a very few cases we received such details where the breeder gave additional information, such as he/she experienced that the puppies’ reaction showed within or between-litter variability to the barking of other dogs, for example.

P6 line 170 – it would helpful to mention that paternal status of other dog(s) was tracked as that comes up later in the analysis.

RESPONSE: We added this detail now, thank you (Line 229).

P6 line 194 – it is not clear to me why you chose to use ordinal regression for the behavioral reactions when there doesn’t appear to be a clear rank order to the behaviors (ex., puppies have no reaction, puppies run away/are quiet (fear), puppies bark (which could be assertive/aggressive response, fear response, playful, etc.,).

RESPONSE: Thank you for this question, we realize that our choice for the statistical method may need some elaboration. We sorted the answers to a rank order based on the ‘intensity’ of the puppies’ response. Nor reaction/ ignorance < getting quiet, hide < join to the barking.

It would be helpful to specify what assumptions were checked for the tests (e.g., test of parallel lines for ordinal logistic regression) and how the models fared in them. Alpha of .05 was used to determine significance, I assume?

RESPONSE: Yes, level of significance (alpha) was 0.05 in each test. The results of testing of parallel lines are the following (in case of ordinal logistic regression):

Puppies’ reaction to adult dogs’ barking

housing method of the puppies (κ2(2)=9.363; P=0.009)

Test of Parallel Linesa

Model

-2 Log Likelihood

Chi-Square

df

Sig.

Null Hypothesis

32,194

General

26,591

5,603

4

,231

timing of separation of puppies (κ2(2)=4.631; P=0.099)

Test of Parallel Linesa

Model

-2 Log Likelihood

Chi-Square

df

Sig.

Null Hypothesis

25,403

General

23,261

2,143

4

,710

aggressive behavior with the puppies (κ2(1)=0.628; P=0.428)

Test of Parallel Linesa

Model

-2 Log Likelihood

Chi-Square

df

Sig.

Null Hypothesis

21,984

General

20,598

1,386

2

,500

FCI breed groups (κ2(2)=1.112; P=0.573)

Test of Parallel Linesa

Model

-2 Log Likelihood

Chi-Square

df

Sig.

Null Hypothesis

35,391

General

18,041

17,350

4

,002

genetically clustered breed groups (κ2(2)=0.598; P=0.742)

Test of Parallel Linesa

Model

-2 Log Likelihood

Chi-Square

df

Sig.

Null Hypothesis

33,586

General

20,366

13,219

4

,010

 Results

In general, it would help the reader if the dependent variables were reported in terms of their actual measurements in the results. A statement like “females dogs under one year of age showed the strongest reaction” to puppies’ whining is difficult to interpret without referring back to the method. A small clarification of what the actual reaction was would make this result more impactful. Figures would be easier to interpret with this information as well (rather than just numbers 1, 2, 3). P6 line 212 – is Pal 2015 a citation? In that case, it needs to be replaced with a citation number.

RESPONSE: We added more detail of the actual behavioral response where the Reviewer requested it. Figures now include more informative title along the vertical axis. Figure captions provide detailed explanation to the horizontal axis categories. We adjusted the mentioned citation to the numbered format.

Discussion

P13 line 312 – this conclusion seems to be rather far-reaching given the small sample size and limited number of countries and breeds included. Well-written discussion which nicely situates this study’s findings in the larger field. While there is some mention of limitations, however, more discussion of specific limitations, and their impact on generalization of results, is warranted. Specifically, although the small sample was addressed (but could use more discussion), the sampling method was not. Given the likely biases that may have resulted from an online convenience sample, especially sourcing from social media, a discussion of the limited number of countries and the possible effect of a non-response bias (what about breeders without (reliable?) internet, or those not on social media, or those that didn’t want to respond due to problematic behaviors in their litters or ‘packs’?) is needed. Additionally, there is just one brief mention of breeders’ interpretation of dog behavior. Given the large number of studies examining people’s/owner’s lack of knowledge or misperceptions about dog behavior, there should be a more extensive discussion of this limitation. This survey is entirely self-report which, while capitalizing on a strength, is also a limitation, particularly related to validity. This also relates to the limited information about the questions and response options mentioned above; there is some research examining how different methods of asking questions about dog behavior can result in different answers; therefore, it is important that the response options be described and then discussed in the discussion.

RESPONSE: We agree with the Reviewer that a more detailed ‘limitations’ section is needed to the Discussion – now we included now a paragraph dedicated to this issue at the beginning of the Discussion chapter (Lines 384-399).

Minor Grammatical Comments:

In US English, “e.g.” is always followed by a comma: “e.g.,” Line 61, “millions” should be singular in this case according to common US usage Line 76, “year” should be plural; and suggest changing “sustain themselves mainly from…” to “by” Line 78, need a closing parenthesis Line 84, comma at the end of the sentence after “environment” is not needed Line 90, need a closing parenthesis (here and elsewhere)

RESPONSE: Thank you for the corrections, we made changes to the manuscript accordingly.
